

# Molecular evidence that the Channel Islands populations of the orange-crowned warbler (*Oreothlypis celata*; Aves: Passeriformes: Parulidae) represent a distinct evolutionary lineage

Zachary R. Hanna[1,2], Carla Cicero[1] and Rauri C.K. Bowie[1,2]

[1] Museum of Vertebrate Zoology, University of California, Berkeley, CA, United States of America
[2] Department of Integrative Biology, University of California, Berkeley, CA, United States of America

## ABSTRACT

We used molecular data to assess the degree of genetic divergence across the breeding range of the orange-crowned warbler (*Oreothlypis celata*) in western North America with particular focus on characterizing the divergence between *O. celata* populations on the mainland of southern California and on the Channel Islands. We obtained sequences of the mitochondrial gene *ND2* and genotypes at ten microsatellite data for 192 *O. celata* from populations spanning all four recognized subspecies. We recovered shallow, but significant, levels of divergence among *O. celata* populations across the species range. Our results suggest that island isolation, subspecies (delineation by morphology, ecological, and life-history characteristics), and isolation-by-distance, in that order, are the variables that best explain the geographic structure detected across the range of *O. celata*. Populations on the Channel Islands were genetically divergent from those on the mainland. We found evidence for greater gene flow from the Channel Islands population to mainland southern California than from the mainland to the islands. We discuss these data in the context of differentiation in phenotypic and ecological characters.

## INTRODUCTION

Oceanic islands have served as a natural laboratory for evolutionary studies for decades (*Crawford, 2012*). Patterns of phenotypic and genetic divergence on islands with varying degrees of isolation shed light on the processes of adaptation and speciation (*Losos & Ricklefs, 2009*; *Greenberg & Danner, 2013*) and provide data for evaluating traits that promote biodiversity (*Lomolino, 2005*; *Gunderson, Mahler & Leal, 2018*). Furthermore, comparisons of island taxa and their mainland counterparts are fundamental to assessing the taxonomic status of island endemics, many of which are of conservation concern (*Wilson et al., 2009*).

The California Channel Islands are well-known for their endemic or near endemic species and subspecies of birds (*Johnson, 1972*; *Jones & Diamond, 1976*). Of the forty-one native

Corresponding author
Rauri C.K. Bowie,
bowie@berkeley.edu

land bird species found on these islands, thirteen (32%) show phenotypic differentiation between the islands and mainland (*Johnson, 1972*). The islands are divided into two groups that differ geologically and biologically: the northern islands (San Miguel, Santa Rosa, Santa Cruz, and Anacapa) and the southern islands (San Nicolas, Santa Barbara, Santa Catalina, and San Clemente). Together, they extend for 260 km off the coast of southern California and range between 20 and 98 km from the mainland (*Schoenherr, Feldmeth & Emerson, 1999*). Patterns and processes of avian (especially passerine) diversification on the Channel Islands have been a topic of interest among ornithologists for decades (*Diamond, 1969*; *Johnson, 1972*; *Lynch & Johnson, 1974*; *Greenberg & Danner, 2013*). Apart from the following, few Channel Islands bird taxa have been the subject of published genetic studies: *Aphelocoma californica* and *A. insularis* (*Delaney, Zafar & Wayne, 2008*); *Melospiza melodia* (*Wilson et al., 2009*); *Lanius ludovicianus*, (*Mundy, Winchell & Woodruff, 1997*; *Caballero & Ashley, 2011*); *Eremophila alpestris* (*Mason et al., 2014*); and *Artemisiospiza belli* (*Karin et al., 2018*). Overall, these studies have shown that the Channel Islands harbor genetic distinctiveness in avian populations and that levels of divergence and gene flow between the islands and mainland vary among taxa.

The orange-crowned warbler (*Oreothlypis celata*) is currently divided into four subspecies that differ in plumage color (Figs. S1 and S2), size, molt patterns, habitat, song, and timing of migration and breeding (*Foster, 1967*; *Dunn & Garrett, 1997*; *Gilbert, Sogge & Van Riper III, 2010*). *Oreothlypis celata celata* (*Say, 1823*) breeds primarily in low, deciduous shrub-dominated thickets in northern North America, including most of Alaska through eastern Canada. *Oreothlypis celata lutescens* (*Ridgway, 1872*) prefers to nest in the dense riparian-chaparral ecotone with vertical structure provided by oaks or conifers along the Pacific coast from southeastern Alaska through California (*Dunn & Garrett, 1997*). *Oreothlypis celata sordida* (*Townsend, 1890*) nests in scrub and woodland on all eight California Channel Islands as well as on the Islas Coronado and Islas de Todos Santos off the northwestern coast of Baja California and in restricted areas on the coast of mainland southern California (*Dunn & Garrett, 1997*; *Schoenherr, Feldmeth & Emerson, 1999*). This subspecies is the only one that predominantly nests off the ground on the mainland (*Gilbert, Sogge & Van Riper III, 2010*). *Oreothlypis celata orestera* (*Oberholser, 1905*) nests in dense riparian areas and, at higher elevations, in stands of aspen groves in the Rocky Mountains from northern British Columbia through southern New Mexico and in some mountain ranges within the western deserts of North America (*Dunn & Garrett, 1997*).

Analyzing the geographic differentiation and distribution patterns of Channel Island birds, *Johnson (1972)* found evidence of both single and multiple colonization events, depending on the particular taxon. For *Oreothlypis celata*, he hypothesized that the insular *O. c. sordida* originated from a single colonization from the mainland to the northern Channel Islands, followed by differentiation and subsequent dispersal among the islands and recolonization of the mainland in areas that were locally unsuitable for *O. c. lutescens*. He also hypothesized that *O. c. sordida* is more closely related to Rocky Mountain *O. c. orestera* populations than to Pacific coast *O. c. lutescens* populations, suggesting a relictual pattern of evolution and distribution.

In the only published genetic study of *Oreothlypis celata*, *Bull et al. (2010)* used mitochondrial DNA (mtDNA) and microsatellite data to assess the relationships between northwestern North American populations of *Oreothlypis celata celata* and *O. c. lutescens* on Haida Gwaii, Canada. They found low, but statistically significant, differentiation between populations, suggesting recent divergence. They also found a pattern consistent with isolation-by-distance. However, because *Bull et al. (2010)* did not include the other two *O. celata* subspecies (*O. c. orestera* and *O. c. sordida*) in their analyses, their data do not provide insight into broader patterns and processes of differentiation across the species, including between Channel Islands and mainland populations.

In order to analyze broad-scale divergences among populations, we sampled mitochondrial and nuclear genetic data from all four subspecies of *Oreothlypis celata*. We assessed the relationship between Channel Island and mainland southern California populations and determined the relative rates of migration between these populations to test (*Johnson, 1972*) hypotheses about the origin and differentiation of *O. celata* on the Channel Islands. We discuss these data in the context of what is known about avian differentiation on the islands.

## MATERIALS AND METHODS

### Population sampling

We obtained blood and/or frozen tissue samples from 192 *Oreothlypis celata* individuals collected between 1983 and 2009 (Table S1) from western North America representing each of the four subspecies (Table S1 and Fig. 1). To control for post-breeding dispersal, we used only samples collected during the breeding months of early April through July (*Gilbert, Sogge & Van Riper III, 2010*). We also obtained frozen tissue samples from two Nashville warblers (*Oreothlypis ruficapilla*) to use as outgroups in our analyses. We obtained samples from museum tissue collections (Table S1) and collected samples under California Department of Fish and Game scientific collecting permit numbers SC-458 and SC-10109, US Fish and Wildlife Service permit number MB153526, and with permission from the UC Berkeley Animal Care and Use Committee under Animal Use Protocols R285 and R317.

We examined populations at several hierarchical levels. First, we analyzed the data using all of the samples without *a priori* groupings. When these initial analyses revealed little spatial structure in the genetic data, we grouped the samples into eight populations (Fig. 1) based on their geographic proximity. This enabled us to explore the extent to which variation across ecosystem boundaries (geography) and present taxonomy (subspecies) are reflected in the genetic data. To explore the importance of geographic variation, we grouped the samples on either side of two separate geographic divisions: northern versus southern (populations 1–3 and 4–8, respectively, in Fig. 1) and coastal versus interior (populations 2, 6–8 and 1, 3–5, respectively, in Fig. 1). Our division between the northern and southern samples near the Pacific Coast fell at the southern limit of the Cascade Range in northern California. In the interior, we divided northern from southern samples between the Canadian Rocky Mountains and the Southern Rocky Mountains at the northern Idaho Clearwater River drainage. These landmarks are ecologically significant as they mark the
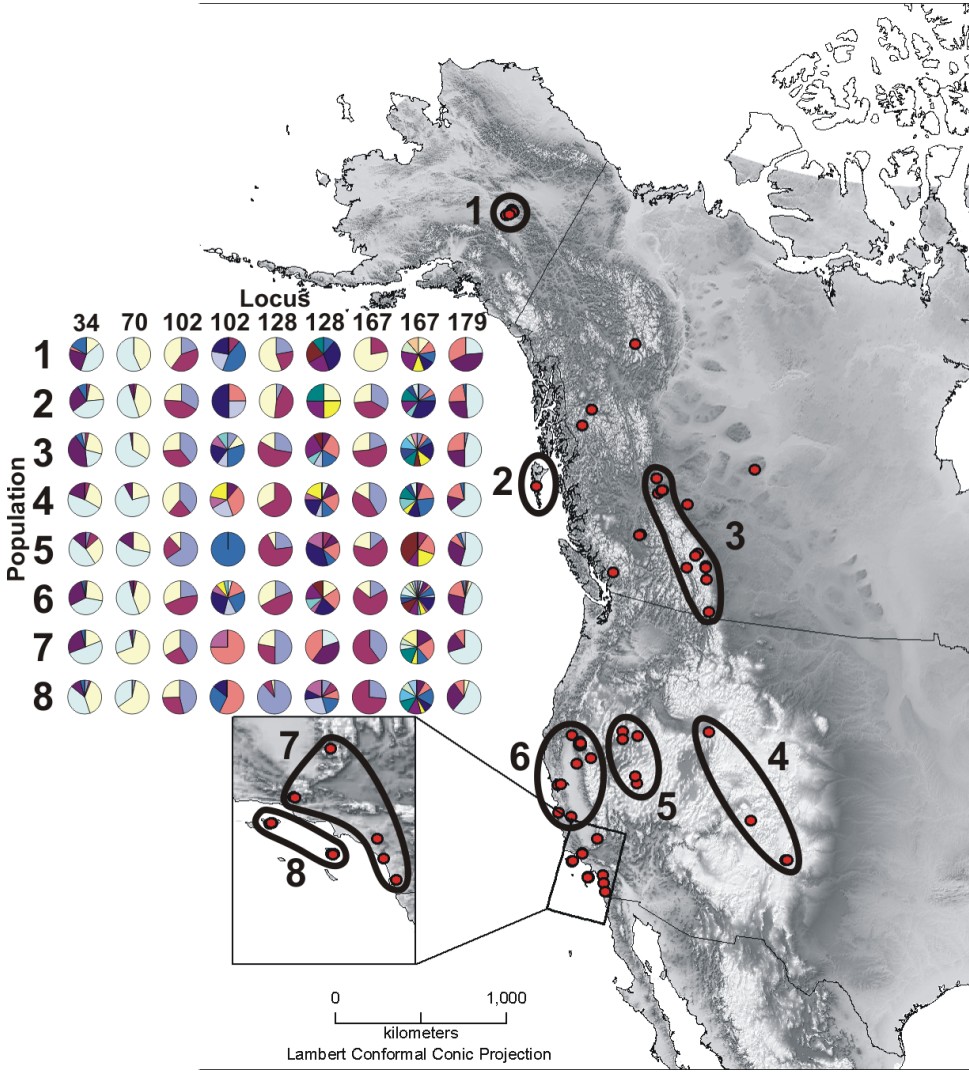

**Figure 1  Sample map and microsatellite allele pie charts.** Depicted here are all *Oreothlypis celata* sampling localities and the associated population designations used in this study. Population numbers correspond to the "Pop #" column in Table 1. We also provide an across-population comparison of the percent prevalence of a subset of the alleles found in our samples for the three most variable (Vce102, Vce128, and Vce167) and three least variable (Vce34, Vce70, and Vce179) loci. For each population, we present the percent prevalence of both the three most common and the rare alleles. We define rare alleles as those whose average occurrence in populations represents less than 5% of the allele pool. Loci Vce70 and Vce102 were exceptions to this definition. Due to the small total number, we included all five detected alleles for Vce70. There were so many rare alleles for Vce102 that we defined the rare alleles for this locus as those with an average population occurrence of <1% of the total allele pool. For the least variable loci, we depict the percent prevalence of the three most common alleles and the rare alleles together in the same pie chart. Due to the large number of rare alleles in the most variable loci, we have depicted the rare alleles in a separate pie chart. For Vce102, Vce128, and Vce167, the left pie charts display the percent prevalence of the three most common alleles and the right graphs represent the percent prevalence of the rare alleles. The prevalence percentages depicted in the pie charts are all relative as the total prevalence of all alleles must sum to one. We recommend that the reader compare graphs vertically, across populations. A given color represents different alleles across columns.

**Table 1  Microchondrial sequence data summary statistics.** This table presents summary statistics for the *ND2* mitochondrial sequence data for each population. We list the number of individuals sampled ($N$) and the number of haplotypes in each population. We provide estimates of haplotype diversity ($h$) with standard deviation, nucleotide diversity ($\pi$) with standard deviation, Tajima's D, Fu's $F_s$, and Harpending's Raggedness Index. The named "North" population includes Pop 1 and 3. The "South" population includes Pop 4 through 7. Values followed by one asterisk are significant with $p < 0.05$ and values followed by two asterisks are significant with $p < 0.001$.

| Pop # | Population | $N$ | Number of haplotypes | $h$ | $\pi$ | Tajima's D | Fu's $F_s$ | Harpending's Raggedness Index |
|---|---|---|---|---|---|---|---|---|
| | North | 42 | 23 | 0.94+/-0.02 | 0.0029+/-0.0017 | −1.80* | −16.50** | 0.023 |
| | South | 92 | 42 | 0.94+/-0.01 | 0.0030+/-0.0017 | −2.35** | −26.42** | 0.018 |
| 1 | Fairbanks | 15 | 9 | 0.89+/-0.06 | 0.0027+/-0.0017 | −1.16 | −3.05* | 0.126 |
| 2 | Haida Gwaii | 20 | 11 | 0.84+/-0.08 | 0.0028+/-0.0017 | −1.73* | −4.04* | 0.063 |
| 3 | Northern Rocky Mtns. | 19 | 11 | 0.89+/-0.06 | 0.0029+/-0.0018 | −1.31 | −4.19* | 0.030 |
| 4 | Southern Rocky Mtns. | 17 | 10 | 0.79+/-0.10 | 0.0019+/-0.0013 | −2.25** | −5.44** | 0.012 |
| 5 | Nevada | 16 | 10 | 0.83+/-0.10 | 0.0020+/-0.0013 | −2.10* | −5.51** | 0.041 |
| 6 | Northern California | 43 | 25 | 0.96+/-0.01 | 0.0037+/-0.0021 | −2.07* | −16.47** | 0.018 |
| 7 | Southern California | 16 | 9 | 0.85+/-0.08 | 0.0028+/-0.0017 | −1.71* | −2.66 | 0.105 |
| 8 | Channel Islands | 30 | 4 | 0.19+/-0.10 | 0.0002+/-0.0003 | −1.73* | −3.38** | 0.417 |

southern extents of cedar-hemlock forest ecosystems (*Brunsfeld et al., 2001*) and have been hypothesized by many as sites of lineage contact in various taxa (*Soltis et al., 1997*; *Swenson & Howard, 2005*; *Burg et al., 2006*). We divided coastal from interior samples by designating as interior all areas east of the Alaska Range, Coast Mountains, the Cascades, and the Sierra Nevada, as splits between coastal and interior populations have been hypothesized in other warbler taxa (*Bermingham et al., 1992*). Finally, we grouped samples based on the four existing subspecific designations. We utilized each of these four separate sample groupings in subsequent analyses.

## Laboratory procedures

We extracted DNA from blood or frozen tissues using a DNeasy Blood & Tissue Kit (Qiagen, Hilden, Germany) following the Qiagen protocol for animal tissues. We sequenced the mitochondrial genes NADH subunit 2 (*ND2*) and ATP Synthase subunit 6 (*ATP6*), both of which are commonly used in avian phylogeographic studies. We amplified a 1041 base pair (bp) fragment of the *ND2* gene using the polymerase chain reaction (PCR) with primers L5204 and H6312 (*Sorenson et al., 1999*). PCR reactions (10 µL) contained 1X PCR Buffer (10 mM Tris-HCl, 1.5 mM MgCl$_2$, 50 mM KCl, pH 8.3), 0.6 µM of each primer, 200 µM of each dNTP, 0.6 U of *Taq* and approximately 5–10 ng of genomic DNA. The PCR profile included an initial denaturation at 94 °C for 2 min; followed by 35 cycles of denaturation at 94 °C for 30 s, annealing at 53 °C for 30 s, and extension at 72 °C for 1 min; with a final extension at 72 °C for 10 min. We amplified a 704 bp fragment of the *ATP6* gene by PCR using the primers a8PWL and C03HMH (*Hunt, Bermingham & Ricklefs, 2001*). The PCR profile followed that for the *ND2* gene, except for annealing at 54 °C and extension for 45 s during the 35 cycle phase before the final extension.

We purified the PCR products using Exonuclease I and Shrimp Alkaline Phosphatase (ExoSAP-IT$^{TM}$; Applied Biosystems, Waltham, MA, USA) and sequenced the purified products using Big Dye terminator chemistry v. 3.1 (Applied Biosystems) and an AB PRISM

3730 DNA Analyzer (Applied Biosystems). We analyzed only samples for which we obtained sequences of both the forward and reverse DNA strands. We aligned complementary DNA strands, edited all sequences, detected stop codons, and aligned consensus sequences using Sequencher version 4.7 (Gene Codes Corporation, Ann Arbor, MI, USA). After obtaining 704 bp of *ATP6* for 106 individuals, we detected the presence of a pseudogene in sequences and thus eliminated the *ATP6* gene from further analyses.

We used ten polymorphic microsatellite markers (Vce34, Vce50, Vce70, Vce102, Vce103, Vce109, Vce116, Vce128, Vce167, and Vce179) developed for *O. celata* (*Bowie et al., 2017*). All ten loci were tetranucleotide repeats and three of them had imperfect core repeats. We amplified these microsatellites using PCR in 10 µL reactions containing: 1x PCR Buffer (10 mM Tris-HCl, 1.5 mM $MgCl_2$, 50 mM KCl, pH 8.3), 0.6 µM of each primer, 200 µM of each dNTP, 0.6 U of *Taq* and approximately 5-10 ng of genomic DNA. The PCR conditions included one denaturation cycle at 94 °C for 2 min and 30 cycles consisting of 15 s of denaturation at 94 °C, 15 s of annealing at 50–55 °C, and 15 s of extension at 72 °C. We used T4 DNA polymerase (New England Biolabs, Ipswich, MA, USA) treatment to clean the PCR products of the Vce34, Vce50, Vce102, Vce103, Vce128, and Vce179 markers (*Ginot et al., 1996*). We mixed the samples with formamide and GS-500 LIZ size standard (Applied Biosystems) and analyzed them using an AB PRISM 3730 DNA Analyzer. We conducted allele binning and genotyping using Genemapper version 4.0 (Applied Biosystems).

## Mitochondrial DNA analyses

We analyzed the *ND2* sequences using maximum likelihood (ML), neighbor-joining (NJ), and maximum parsimony (MP) algorithms. We used RAxML BlackBox (*Stamatakis, Hoover & Rougemont, 2008*) to construct an ML tree with 100 bootstrap replicates and PAUP* version 4.0b10 (*Swofford, 2003*) to construct NJ and MP trees. Preliminary analyses of the mtDNA data using NJ, ML, and MP algorithms were not informative and intraspecific datasets often do not comply with the assumptions of MP and ML algorithms (*Posada & Crandall, 2001*). Therefore, we did not further explore tree-building methods that assume bifurcation of lineages by default and instead focused on the population genetics approaches described hereafter.

We generated a statistical parsimony network using TCS version 1.01 (*Clement, Posada & Crandall, 2000*) to visualize relationships among haplotypes and to analyze phylogeographic structure. In addition, we used analysis of molecular variance (AMOVA) in Arlequin version 3.1 (*Excoffier, Smouse & Quattro, 1992*; *Excoffier, Laval & Schneider, 2007*) to calculate the proportion of total mtDNA genetic variation explained by population groupings. The AMOVA provided estimates of overall $F_{ST}$ and its analogue, $\Phi_{ST}$ (calculated using the Tamura-Nei model with a 0.05 gamma correction), using a non-parametric permutation approach to determine significance levels (*Excoffier, Smouse & Quattro, 1992*). We used Arlequin version 3.1 to examine genetic structure among population subdivisions by calculating pairwise $F_{ST}$ and $\Phi_{ST}$ statistics (10,000 permutations) and applying sequential Bonferroni corrections when evaluating significance (*Rice, 1989*). We also used Arlequin version 3.1 to estimate haplotype diversity (*h*) and nucleotide diversity ($\pi$) (*Nei, 1987*), to calculate pairwise mismatch distributions for populations (Sum of Squared deviations

and Harpending's Raggedness index calculated to test goodness of fit; 10,000 bootstrap replicates), and to run two tests of selective neutrality, Tajima's D (*Tajima, 1989*) and Fu's *F* (*Fu, 1997*) tests.

We performed a spatial analysis of molecular variance (SAMOVA) using SAMOVA 1.0 (*Dupanloup, Schneider & Excoffier, 2002*) to assess the geographic arrangement of genetic structure. Unlike an AMOVA, this method does not require an a priori definition of populations. Instead, it uses sequence and geographic coordinate data (Lambert projection) to maximize the proportion of total genetic variation among populations (*Dupanloup, Schneider & Excoffier, 2002*). We identified the most likely partitioning of the samples by running SAMOVA 1.0 repeatedly with 2 to 20 groups and looking for the division assemblage with a maximized $F_{CT}$ (*Dupanloup, Schneider & Excoffier, 2002*).

## Microsatellite analyses

We used Arlequin version 3.1 (*Excoffier, Laval & Schneider, 2007*) to calculate observed ($H_O$) and expected ($H_E$) heterozygosity values. We tested for Hardy–Weinberg equilibrium (HWE) and heterozygote deficiency using Genepop version 4.0.10 (10,000 dememorization steps, 1,000 batches, 10,000 iterations) (*Raymond & Rousset, 1995*; *Rousset, 2008*). In addition, we tested the microsatellite genotypes in each population and at each locus for linkage equilibrium using Genepop version 4.0.10 (10,000 dememorization steps, 1,000 batches, 10,000 iterations) (*Raymond & Rousset, 1995*), applying sequential Bonferroni corrections when evaluating significance (*Rice, 1989*). We examined null allele presence using Micro-Checker version 2.2.3 (*Van Oosterhout et al., 2004*) and used FSTAT version 2.9.3.2 (*Goudet, 1995*; *Goudet, 2001*) to estimate allelic richness ($R_s$), which controls for sample size when comparing the number of alleles among populations (*Leberg, 2002*).

We tested the proportion of total genetic variance explained by population groupings by performing an AMOVA (*Excoffier, Smouse & Quattro, 1992*) in Arlequin version 3.1, which provided estimates of overall $F_{ST}$. We calculated the significance levels for the AMOVA using a non-parametric permutation approach (10,000 permutations) (*Excoffier, Smouse & Quattro, 1992*). We examined genetic structure among population subdivisions by calculating pairwise $F_{ST}$ values using Arlequin version 3.1 (10,000 permutations) and pairwise $R_{ST}$ values using RSTCALC version 2.2 (*Goodman, 1997*), applying sequential Bonferroni corrections for multiple simultaneous comparisons when evaluating significance (*Rice, 1989*).

We tested the pairwise correlation between direct geographic and genetic (*Nei, 1972*) distances (isolation-by-distance) among all individuals sampled by conducting a Mantel test using GenAlEx version 6.1 (*Peakall & Smouse, 2006*; *Peakall & Smouse, 2012*). We also used GenAlEx version 6.1 to run a principal coordinates analysis (PCA) in order to examine the organization of the genetic structure.

In a further effort to detect spatial organization in our sample assemblage, we analyzed our dataset of ten microsatellite loci using Structure version 2.3.4 (*Pritchard, Stephens & Donnelly, 2000*; *Falush, Stephens & Pritchard, 2003*; *Hubisz et al., 2009*; *Pritchard, Falush & Hubisz, 2012*). This method uses Bayesian clustering to examine genetic frequencies across loci and attempts to identify the number of clusters ($K$)

based on the likelihood values for varying $K$ values. We performed preliminary analyses without providing any information concerning population designations. After these initial analyses, we then designated eight populations in the input and used this information as a prior (LOCPRIOR) (*Hubisz et al., 2009*) in further analyses to improve population discrimination. We implemented the analyses using the admixture model with correlated allele frequencies (*Falush, Stephens & Pritchard, 2003*), examined $K = 1 - 20$, executed a 100,000 MCMC iteration burn-in, and then performed 1,000,000 subsequent MCMC iterations. We replicated the simulation at each $K$ twenty times. To assist in identifying the optimal $K$, we used Structure Harvester version 0.6.94 (*Earl & VonHoldt, 2012*; *Earl, 2014*), which uses the *Evanno, Regnaut & Goudet (2005)* method to identify the number of clusters. We ran Structure and Structure Harvester using StrAuto version 1.0 (*Chhatre & Emerson, 2017*; *Chhatre & Emerson, 2018*) with GNU Parallel version 20141022 (*Tange, 2011*). To align clusters across the Structure runs, we ran CLUMPP version 1.1.2 (*Jakobsson & Rosenberg, 2007*) and then used a modified version of Distruct version 2.2 (*Raj, Stephens & Pritchard, 2014*; *Chhatre, 2016*; *Hanna, Cicero & Bowie, 2018*) to plot the clusters.

Based on the results of the Structure analysis described above, we ran two additional Structure analyses to check for the presence of substructure. We first analyzed the Channel Islands samples with the samples from Santa Cruz Island and Santa Catalina Island split into separate populations. We used the parameters as detailed above, including the LOCPRIOR for $K = 1 - 10$. We then analyzed the seven remaining populations with the same parameters as above for $K = 1 - 20$.

In order to assess the relative rate of migration between the Channel Islands and mainland southern California, we ran IMa2p version 58a0260 (*Sethuraman & Hey, 2015*; *Sethuraman, 2017*). We input both the *ND2* sequences and microsatellite genotypes and performed three separate runs each with 15 chains, 1,000,000 burnin steps, and 2,000,000 further steps following the burnin. We have provided further methodology details in ocwa-popgen version 1.0.0 on GitHub (*Hanna, Cicero & Bowie, 2018*).

## RESULTS

### mtDNA sequence variation

We obtained a complete 1041 bp fragment of the mtDNA *ND2* gene for 192 *Oreothlypis celata* and two *O. ruficapilla* individuals; there were no missing data and no insertions, deletions, or gaps. After merging identical sequences, we found 72 unique haplotypes (Table S1) with 81 variable sites. We found no evidence for selection ($P = 0.702$) between *Oreothlypis celata* sequences and two sequences of the closely related *O. ruficapilla* (*Lovette, Bermingham & Sheldon, 2002*).

### mtDNA haplotype network

Examination of the statistical parsimony network revealed shared alleles, regardless of how we grouped samples into populations (Fig. 2 and Fig. S3). The haplotypes clustered largely along a north-south geographic axis, but the majority of the Haida Gwaii *Oreothlypis celata* possessed haplotypes in the "southern" group. Three mutational differences separate

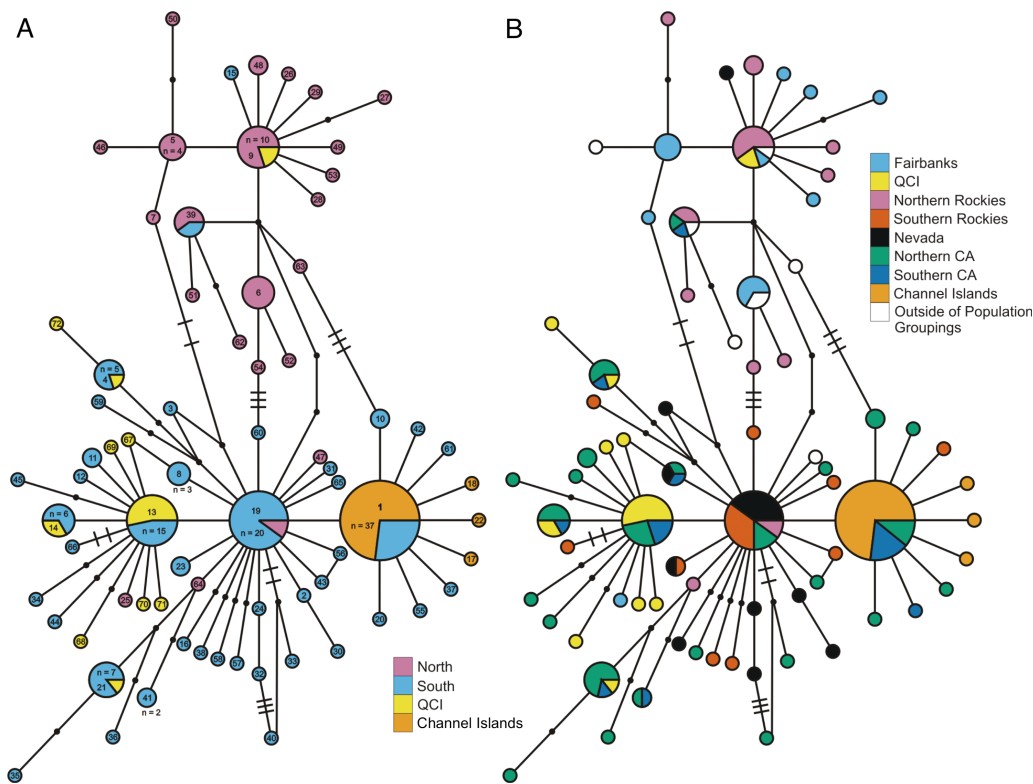

**Figure 2 *ND2* haplotype network.** (A) depicts the *ND2* haplotype network shaded according to samples' designation in the northern or southern population. (B) is the haplotype network shaded using sample assignment under the eight population grouping arrangement, a more fine-scale partitioning than the north-south grouping schema. The haplotype numbers in (A) correspond with the numbers in Table S1. Circle sizes are proportional to the number of individuals with each haplotype. Lines connect haplotypes that differ by one mutation. Dots represent inferred haplotypes. Hash marks indicate the number of mutations between haplotypes separated by more than one mutational difference. For one circle of each size, we have labeled the number of individuals represented by that circle following "*n* =".

the major haplotype clusters of the northern and southern *O. celata* with some outlier individuals falling into each grouping.

The *Oreothlypis celata* haplotypes from the Channel Islands clustered much more tightly than those from Haida Gwaii. We found four *ND2* haplotypes among the Channel Islands *O. c. sordida*, but the majority of individuals shared a single haplotype; the three other Channel Islands haplotypes appeared only in one individual each (Fig. 2). There was at most one mutational difference between the haplotype of a Channel Islands *O. celata* and the next Channel Islands haplotype. Although we found three singleton, private Channel Islands *ND2* haplotypes, individuals from northern and southern California shared the most common Channel Islands haplotype. The Haida Gwaii samples, with eleven haplotypes, were more loosely clustered than the Channel Islands samples with a maximum of nine mutational steps between individuals (Fig. S3).

When we identified samples by subspecies (Fig. 3), we found no interior *Oreothlypis celata orestera* individuals that shared haplotypes with the Channel Islands *O. c. sordida*.

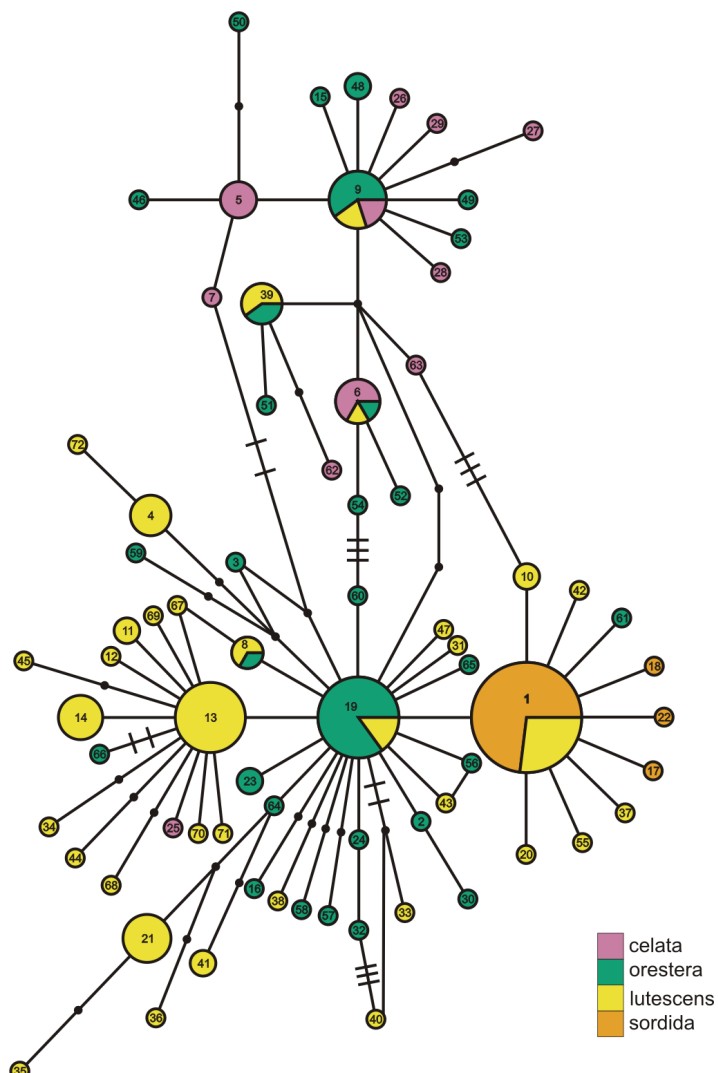

**Figure 3** *ND2* **haplotype network with subspecies population grouping.** This is the *ND2* haplotype network colored by the subspecies designations of samples. The haplotype numbers correspond with the numbers in Table S1. The size of each circle is proportional to the number of individuals with that haplotype. Lines connect haplotypes that differ by one mutation. Dots represent inferred haplotypes. Hash marks indicate the number of mutations between haplotypes separated by more than one mutation.

We did, however, find *O. c. orestera* haplotypes that were one mutational step away from *O. c. sordida* haplotypes (Fig. 3). The main cluster of *O. c. lutescens* haplotype diversity was separated from the *O. c. sordida* haplotype cluster by a haplotype more often found in *O. c. orestera* than in *O. c. lutescens*. The haplotypes did not appear to cluster across a coast-interior axis (Fig. S3). However, with the exception of one Haida Gwaii haplotype, the island populations of the Channel Islands and Haida Gwaii did not share haplotypes with any individuals from interior populations.

**Table 2  Population pairwise divergence statistics.** This table presents divergence statistics for pairwise population comparisons calculated using the *ND2* mitochondrial DNA sequence ($\phi_{ST}$ above diagonal) and microsatellite data ($R_{ST}$ below diagonal). Values followed by asterisks are significant after applying a Bonferroni correction ($p < 0.002$). See Table S1 for the samples included in each population.

| | Fairbanks | Haida Gwaii | Northern Rocky Mountains | Southern Rocky Mountains | Nevada | Northern California | Southern California | Channel Islands |
|---|---|---|---|---|---|---|---|---|
| Fairbanks | – | 0.525* | 0.011 | 0.584* | 0.532* | 0.486* | 0.528* | 0.809* |
| Haida Gwaii | 0.029 | – | 0.481* | 0.152* | 0.166* | 0.061 | 0.110 | 0.564* |
| Northern Rocky Mountains | 0.005 | 0.002 | – | 0.521* | 0.467* | 0.440* | 0.472* | 0.754* |
| Southern Rocky Mountains | 0.119* | 0.087* | 0.026 | – | 0.006 | 0.016 | 0.047 | 0.531* |
| Nevada | 0.141* | 0.092* | 0.033 | 0.000 | – | 0.035 | 0.069 | 0.558* |
| Northern California | 0.027 | 0.020 | 0.000 | 0.038* | 0.054* | – | 0.000 | 0.245* |
| Southern California | 0.103* | 0.079* | 0.025 | 0.089* | 0.095* | 0.040 | – | 0.261* |
| Channel Islands | 0.221* | 0.177* | 0.094* | 0.145* | 0.123* | 0.111* | 0.027 | – |

## Population structure inferred from mtDNA

Variability in mtDNA sequences differed among populations (Table 1). We found that the Channel Islands population had the lowest nucleotide diversity ($0.2 \times 10^{-3}$) of all eight populations, whereas the northern California population had the highest ($3.7 \times 10^{-3}$). The nucleotide diversity of the Haida Gwaii population ($2.8 \times 10^{-3}$) was substantially higher than that of the Channel Islands populations and equaled that of the southern California population ($2.8 \times 10^{-3}$). When grouped into northern and southern population clusters, the two groupings contained almost exactly the same nucleotide diversities ($2.9 \times 10^{-3}$ and $3.0 \times 10^{-3}$, respectively).

Although the statistical parsimony networks (Figs. 2, 3 and S3) did not display evidence of reciprocal monophyly among populations or subspecies, the AMOVA revealed significant differentiation in haplotype frequencies for each of the four alternative groupings of our samples. Overall $F_{ST}$ estimates from our AMOVA analysis of *ND2* sequences were all highly significant ($p < 0.01$) for samples grouped into: (1) northern and southern clusters (0.191); (2) eight populations (0.202); (3) coastal and interior clusters (0.186); and (4) subspecies (0.195). Overall $\Phi_{ST}$ estimates were greater than the $F_{ST}$ estimates for the different population data sets, and were also all highly significant with $p < 0.01$: (1) northern-southern (0.429); (2) eight-population (0.365); (3) coastal-interior (0.254); and (4) subspecies (0.299). The pairwise population $F_{ST}$ values reflected patterns that were nearly congruent to the pairwise $\Phi_{ST}$ estimates, so we have chosen to present only the pairwise $\Phi_{ST}$ estimates (Tables 2–4).

Pairwise population $F_{ST}$ and $\Phi_{ST}$ estimates (0.036 and 0.000, respectively) between Santa Cruz Island (northern Channel Islands) and Santa Catalina Island (southern Channel Islands) were not significant. However, pairwise $\Phi_{ST}$ estimates supported the collective Channel Islands as a distinct population. Pairwise $\Phi_{ST}$ values between the Channel Islands and every other population were significant, ranging from 0.245 to 0.809 with the samples grouped into eight populations (Table 2) and from 0.228 to 0.681 with the samples grouped into northern and southern clusters (Table 3). With the samples grouped into eight populations, we estimated the highest pairwise $\Phi_{ST}$ values between the Channel

**Table 3** **Pairwise divergence statistics of the north, south, and island populations.** We here present the results of pairwise population comparisons with *ND2* mitochondrial DNA sequence ($\phi_{ST}$ above diagonal) and microsatellite ($R_{ST}$ below diagonal) data. Values followed by asterisks are significant after applying a Bonferroni correction ($p < 0.008$).

|  | North | South | Haida Gwaii | Channel Islands |
|---|---|---|---|---|
| North | – | 0.479* | 0.492* | 0.681* |
| South | 0.011 | – | 0.094* | 0.228* |
| Haida Gwaii | 0.013 | 0.038* | – | 0.564* |
| Channel Islands | 0.130* | 0.091* | 0.178* | – |

**Table 4** **Subspecies pairwise divergence statistics.** This table presents divergence statistics for pairwise subspecies comparisons calculated using *ND2* mitochondrial DNA sequence ($\phi_{ST}$ above diagonal) and microsatellite data ($R_{ST}$ below diagonal). Values followed by asterisks are significant after applying a Bonferroni correction ($p \leq 0.008$). See Table S1 for the samples included in each population.

|  | Lutescens | Orestera | Celata | Sordida |
|---|---|---|---|---|
| *lutescens* | – | 0.126* | 0.469* | 0.232* |
| *orestera* | 0.021* | – | 0.258* | 0.375* |
| *celata* | 0.016 | 0.046* | – | 0.786* |
| *sordida* | 0.106* | 0.105* | 0.187* | – |

Islands and the two northern, interior populations (Fairbanks, 0.809; Northern Rocky Mountains, 0.754). Of all of the pairwise comparisons involving the Channel Islands, we estimated the lowest $\Phi_{ST}$ between the Channel Islands and the northern and southern California populations (0.245 and 0.261, respectively).

Pairwise $\Phi_{ST}$ estimates between Haida Gwaii and every other population within the set of eight populations were significant, except for those between Haida Gwaii and the northern and southern California populations. Of all of the Haida Gwaii pairwise comparisons, pairwise $\Phi_{ST}$ was highest (0.564) between the Haida Gwaii and Channel Islands populations. The pairwise $\Phi_{ST}$ estimate was significant between the northern and southern populations (0.479; Table 3), but it was not as high as the estimate between the northern and Haida Gwaii populations (0.492). In contrast, pairwise $\Phi_{ST}$ was much lower between Haida Gwaii and the southern population (0.094; Table 3).

With the samples grouped by subspecies (Table 4), we estimated significant pairwise $\Phi_{ST}$ between *Oreothlypis c. sordida* and all other subspecies, with the lowest values between *O. c. sordida* and *O. c. lutescens* (0.232) and the highest between *O. c. sordida* and *O. c. celata* (0.786). *Oreothlypis c. lutescens* and *O. c. orestera* had the lowest pairwise $\Phi_{ST}$ value of all of the subspecies comparisons. All of the pairwise $\Phi_{ST}$ estimates were significant when we grouped the samples by subspecies (Table 4) and by coastal versus interior populations (Table S2).

## SAMOVA

As we found with our maximum parsimony and maximum likelihood analyses, our SAMOVA analyses indicated that deep genetic structure is not present in our mitochondrial sequence data set. We never obtained a maximized $F_{CT}$ with the SAMOVA analyses, so we

could not reject panmixia or obtain support for population structure greater than $K = 1$. SAMOVA is known to perform poorly in the presence of isolation-by-distance (*Dupanloup, Schneider & Excoffier, 2002*) and we recovered significant isolation-by-distance in the microsatellite data. However, the trend was weak and likely did not greatly affect the SAMOVA analyses. Although we never recovered a maximized $F_{CT}$ with the SAMOVA analyses, we examined the groupings created for $K = 2 - 4$ to see whether the analyses recovered any divisions between northern, southern, and island samples. These analyses partitioned the samples in general agreement with our northern and southern sample groupings. For $K = 2$, we recovered one group composed entirely of northern samples. The second group included all of the southern, Channel Islands, and Haida Gwaii samples as well as samples from five coastal and interior localities (in British Columbia, Alberta, and Fairbanks) in our designated northern population. Grouping samples with $K = 3$ and $K = 4$ created partitions within the northern and southern populations but were not consistent with subspecies boundaries.

## Mismatch distributions

Mismatch profiles that follow a Poisson distribution suggest population growth following an event such as a range expansion (*Rogers & Harpending, 1992*; *Harpending et al., 1993*). Multimodal mismatch profiles can suggest a number of different population dynamic scenarios, such as constant size (*Slatkin & Hudson, 1991*; *Rogers & Harpending, 1992*; *Harpending et al., 1998*), expanding fronts (*Liebers, Helbig & De Knijff, 2001*), and geographic structuring resulting from restricted gene flow (*Marjoram & Donnelly, 1994*). All populations had negative Tajima and Fu statistics and all were statistically significant with the exception of the Fairbanks and Northern Rocky Mountains populations for Tajima's D and the southern California population for Fu's F (Table 1). Harpending's Raggedness indices were not statistically significant for mismatch distributions in any of the populations, indicating that we could not reject a population expansion hypothesis (Table 1). The northern, southern, and Channel Islands populations displayed mismatch profiles following a Poisson distribution, suggesting recent population growth (Fig. 4). With the samples grouped into eight populations, we observed mismatch profiles with a Poisson distribution in all populations except the Fairbanks and Haida Gwaii populations, both of which appeared to have multimodal mismatch profiles (Fig. 4).

## Population structure inferred from microsatellite data

We successfully obtained genotypes for 192 *Oreothlypis celata* individuals at ten microsatellite loci with no missing data apart from three individuals for which we were unable to genotype a subset of the loci (Table 5, Figs. S3 and S4). We found no evidence for null alleles in any microsatellite locus in any population. In addition, there was no evidence for linkage disequilibrium in the northern, southern, Channel Islands, or Haida Gwaii populations; no disequilibrium tests were significant after we applied Bonferroni corrections. We did not observe deviation of observed heterozygosity from Hardy-Weinberg equilibrium (HWE) expectations repeatedly across loci in any of the populations resulting from our various methods of sample grouping. Observed heterozygosity at all

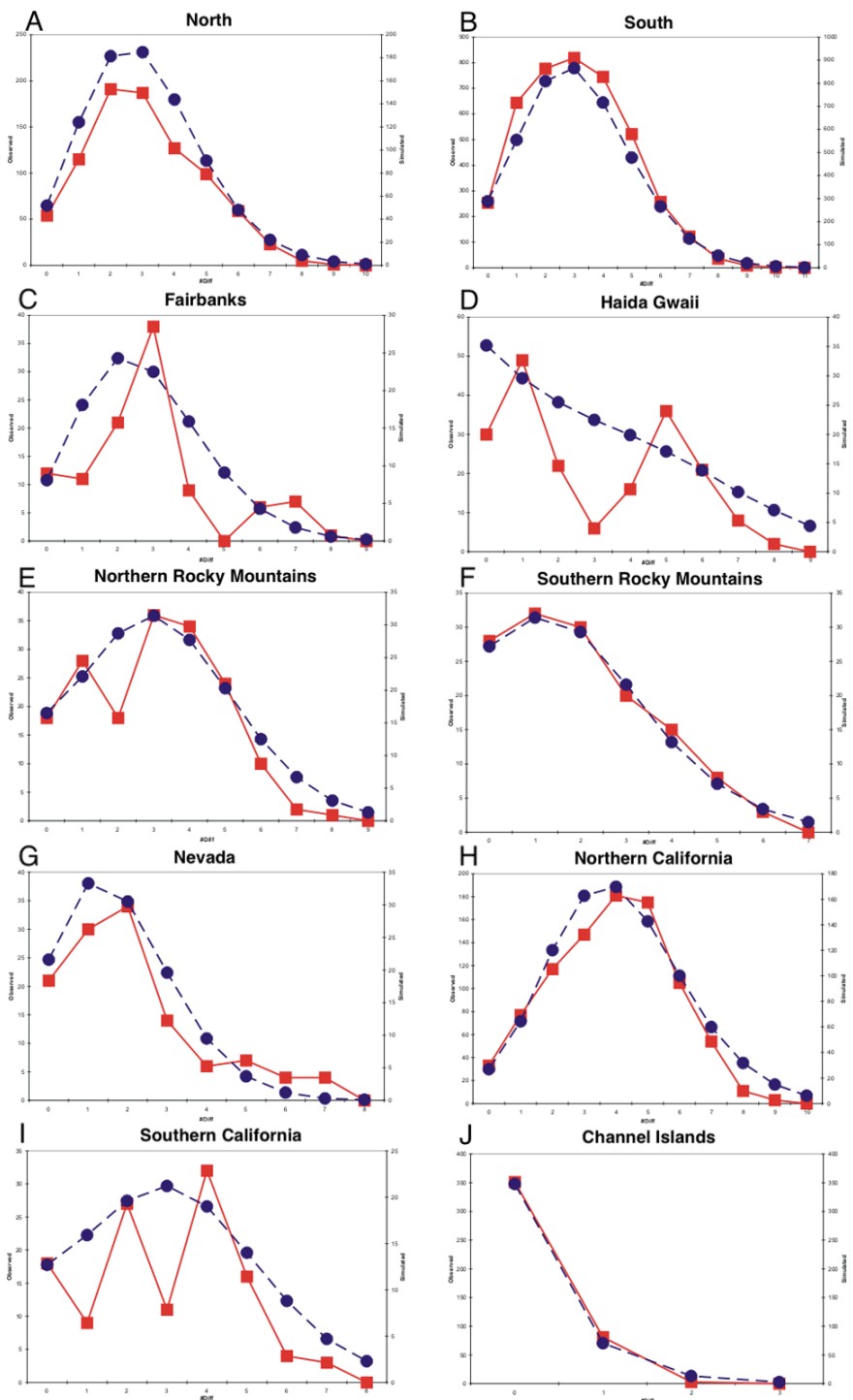

**Figure 4  Mismatch distributions.** (A–J) are mismatch distributions for ten populations. Square points connected by smooth lines represent observed distributions. Circular points connected by dotted lines represent expected distributions for a growing population with the same mean.

ten loci did not differ from that expected under HWE for the northern, southern, Channel Island, and Haida Gwaii population set. However, locus Vce34 was out of HWE in the Fairbanks population, locus Vce167 was out of HWE in the interior population, and locus Vce34 was out of HWE in the *O. c. celata* population.

The overall $F_{ST}$ estimates from our analysis of microsatellite genotypes for the northern-southern, eight-population, coastal-interior, and subspecies population sets (0.017, 0.022, 0.016, 0.020, respectively) were all highly significant ($p < 0.001$). Overall $R_{ST}$ estimates were also highly significant ($p < 0.001$), exhibiting the same pattern as the $F_{ST}$ estimates and exceeded these for the northern-southern, eight-population, coastal-interior, and subspecies population sets (0.055, 0.068, 0.053, 0.058, respectively).

Both the pairwise $F_{ST}$ and $R_{ST}$ estimates from our microsatellite data displayed patterns almost congruent to the pairwise $F_{ST}$ and $\Phi_{ST}$ estimates obtained from the mtDNA data. As with the pairwise $F_{ST}$ and $\Phi_{ST}$ estimates for the mtDNA data, the pairwise population $F_{ST}$ values were smaller than and showed patterns similar to the pairwise $R_{ST}$ estimates, so we chose to present only pairwise $R_{ST}$ estimates (Tables 2–4, and Table S3) here. As with the pairwise $\Phi_{ST}$ estimates, the pairwise $R_{ST}$ estimates supported the existence of a distinct Channel Islands population. In further agreement with the mtDNA analyses, the pairwise $F_{ST}$ and $R_{ST}$ estimates between Santa Cruz Island and Santa Catalina Island (representing the northern and southern Channel Islands, respectively) were not statistically significant. Pairwise $R_{ST}$ estimates between the Channel Islands population and the northern, southern, and Haida Gwaii populations were significant at 0.130, 0.091, and 0.178, respectively (Table 3). When we grouped samples into eight populations, pairwise $R_{ST}$ values between the Channel Islands and every other population, except for southern California, were significant, ranging from 0.027 to 0.221 (Table 2). Within the set of eight populations, the highest pairwise $R_{ST}$ estimate (0.221) was between the Channel Islands and Fairbanks populations. Of the pairwise comparisons amongst the set of eight populations that included the Channel Islands, the lowest pairwise $R_{ST}$ estimate (0.027) was between the Channel Islands and southern California populations; the second-lowest estimate (0.094) was between the Channel Islands and Northern Rocky Mountains. Across all loci, we identified three private alleles in the Channel Islands and four in Haida Gwaii, whereas we found only two private alleles in the southern California population (Table S3). When we grouped samples by subspecies, the highest of the pairwise $R_{ST}$ estimates involving *O. c. sordida* (0.187) was between *O. c. sordida* and *O. c. celata*. The lowest of these estimates (0.105) was between *O. c. sordida* and *O. c. orestera*, but the estimate between *O. c. sordida* and *O. c. lutescens* (0.106) was very close (Table 2).

When we grouped samples into northern, southern, Channel Islands, and Haida Gwaii populations, we found that the pairwise $R_{ST}$ estimates between Haida Gwaii and the southern population were significant, but we did not find significance between Haida Gwaii and the northern population nor between the northern and southern populations. When we grouped the samples into eight populations, the pairwise $R_{ST}$ estimates involving the Haida Gwaii population ranged from 0.002 with the Northern Rocky Mountains to 0.177 with the Channel Islands; estimates were significant with all populations, except for Fairbanks, the Northern Rocky Mountains, and northern California (Table 2). The pairwise

**Table 5** **Variability of the microsatellite loci in the north, south, and island populations.** This table presents the variability of the ten microsatellite loci in each of the four *Oreothlypis celata* populations in the North-South population schema. We indicate the number of individuals genotyped for each locus, "N". Column "A" provides the number of alleles at each locus in each population, with the number of private alleles given in parentheses. We also provide estimated values of allelic richness "$R_S$", observed heterozygosity "$H_O$", expected heterozygosity "$H_E$", and the associated *p*-values for each locus in each population. No *p*-values were significant after Bonferroni correction ($p < 0.005$).

| Population | Locus | N | A (Private Alleles) | $R_S$ | $H_O$ | $H_E$ | *p-val* |
|---|---|---|---|---|---|---|---|
| **North** | | | | | | | |
| | Vce34 | 43 | 10 (1) | 8.51 | 0.698 | 0.827 | 0.085 |
| | Vce50 | 43 | 37 (4) | 24.14 | 0.953 | 0.966 | 0.738 |
| | Vce70 | 42 | 4 (0) | 3.66 | 0.452 | 0.555 | 0.177 |
| | Vce102 | 42 | 12 (3) | 9.664 | 0.738 | 0.836 | 0.223 |
| | Vce103 | 42 | 8 (0) | 7.011 | 0.571 | 0.640 | 0.059 |
| | Vce109 | 42 | 10 (1) | 8.331 | 0.833 | 0.833 | 0.381 |
| | Vce116 | 42 | 10 (2) | 8.146 | 0.786 | 0.839 | 0.331 |
| | Vce128 | 42 | 18 (0) | 14.43 | 0.857 | 0.924 | 0.211 |
| | Vce167 | 43 | 23 (4) | 17.00 | 0.814 | 0.915 | 0.097 |
| | Vce179 | 43 | 8 (0) | 6.869 | 0.860 | 0.788 | 0.190 |
| **South** | | | | | | | |
| | Vce34 | 94 | 10 (0) | 7.428 | 0.798 | 0.804 | 0.633 |
| | Vce50 | 93 | 42 (7) | 21.90 | 0.946 | 0.962 | 0.029 |
| | Vce70 | 94 | 5 (0) | 3.711 | 0.500 | 0.571 | 0.643 |
| | Vce102 | 94 | 13 (3) | 9.119 | 0.766 | 0.827 | 0.250 |
| | Vce103 | 94 | 12 (3) | 6.884 | 0.574 | 0.634 | 0.067 |
| | Vce109 | 94 | 14 (3) | 8.691 | 0.840 | 0.810 | 0.070 |
| | Vce116 | 94 | 10 (1) | 7.471 | 0.830 | 0.813 | 0.829 |
| | Vce128 | 94 | 20 (0) | 13.31 | 0.883 | 0.892 | 0.141 |
| | Vce167 | 94 | 28 (6) | 17.08 | 0.872 | 0.914 | 0.569 |
| | Vce179 | 94 | 11 (2) | 7.593 | 0.840 | 0.779 | 0.424 |
| **Haida Gwaii** | | | | | | | |
| | Vce34 | 20 | 7 (0) | 7.000 | 0.650 | 0.797 | 0.277 |
| | Vce50 | 20 | 24 (3) | 24.00 | 1.000 | 0.962 | 1.000 |
| | Vce70 | 20 | 4 (0) | 4.000 | 0.550 | 0.581 | 0.141 |
| | Vce102 | 20 | 7 (0) | 7.000 | 0.550 | 0.772 | 0.064 |
| | Vce103 | 20 | 8 (0) | 8.000 | 0.650 | 0.669 | 0.899 |
| | Vce109 | 20 | 9 (0) | 9.000 | 0.700 | 0.853 | 0.029 |
| | Vce116 | 20 | 7 (1) | 7.000 | 0.900 | 0.792 | 0.224 |
| | Vce128 | 20 | 9 (0) | 9.000 | 0.750 | 0.768 | 0.493 |
| | Vce167 | 20 | 16 (0) | 16.00 | 0.900 | 0.935 | 0.813 |
| | Vce179 | 20 | 7 (0) | 7.000 | 0.650 | 0.740 | 0.271 |
| **Channel Islands** | | | | | | | |
| | Vce34 | 30 | 9 (0) | 8.308 | 0.700 | 0.779 | 0.364 |
| | Vce50 | 30 | 23 (2) | 19.18 | 0.900 | 0.945 | 0.562 |

**Table 5** (*continued*)

| Population | Locus | N | A (Private Alleles) | $R_S$ | $H_O$ | $H_E$ | *p-val* |
|---|---|---|---|---|---|---|---|
| | Vce70 | 30 | 3 (0) | 2.893 | 0.500 | 0.505 | 0.725 |
| | Vce102 | 30 | 7 (0) | 6.549 | 0.800 | 0.802 | 0.375 |
| | Vce103 | 30 | 2 (0) | 2.000 | 0.467 | 0.364 | 0.295 |
| | Vce109 | 30 | 8 (0) | 7.678 | 0.833 | 0.829 | 0.650 |
| | Vce116 | 30 | 8 (0) | 7.409 | 0.700 | 0.789 | 0.144 |
| | Vce128 | 29 | 14 (1) | 11.85 | 0.724 | 0.704 | 0.467 |
| | Vce167 | 29 | 15 (0) | 13.82 | 0.655 | 0.902 | 0.006 |
| | Vce179 | 30 | 7 (0) | 6.549 | 0.833 | 0.775 | 0.820 |

$R_{ST}$ estimates did not suggest much differentiation within the northern populations, as none of the pairwise $R_{ST}$ estimates involving the Fairbanks, Haida Gwaii, Northern Rocky Mountains, and northern California populations were statistically significant (Table 2). The insignificant pairwise $R_{ST}$ estimate between the Southern Rocky Mountains and Nevada suggested a connection between these populations; pairwise $R_{ST}$ estimates between them and all other populations, except for the Northern Rocky Mountains, were significant (Table 2).

Overall, the microsatellite data revealed little genetic structure and low divergence of populations among our *Oreothlypis celata* samples. Our PCA analysis did not reveal distinct clustering of the samples by population. Mantel tests utilizing geographic distance (GGD) and Log(1+GGD) versus genetic distance (GD) resulted in weak, statistically significant, positive correlation between geographical distance of *O. celata* sampling localities and genetic distance measured at microsatellite loci ($r^2 = 0.015$, $P = 0.006$ for GGD vs. GD and $r^2 = 0.031$, $P = 0.001$ for Log(1+GGD) vs. GD). Our preliminary Structure analyses, in which we did not provide any a priori population information, suggested $K = 1$ as the optimal number of genetic clusters. When we grouped the samples into eight pre-designated populations, the mean ln Pr(X|K) and $\Delta K$ (*Evanno, Regnaut & Goudet, 2005*) suggested $K = 2$ as the optimal number of genetic clusters (Fig. 5). All of the Channel Islands samples had high ancestry (>83%) in one of the clusters, whereas the northernmost samples had the highest ancestry in the other cluster. In our analysis of substructure within the seven populations other than the Channel Islands, $\Delta K$ suggested $K = 2$ as optimal, but the highest mean ln Pr(X | K) was for $K = 1$, although the log probability for $K = 2$ was very similar. With $K = 2$, the southern California, Nevada, and Southern Rocky Mountains populations had high ancestry in one of the clusters and the northern California, Northern Rocky Mountains, Haida Gwaii, and Fairbanks populations had similarly high ancestry in the other genetic cluster (Fig. S4). In our analysis of substructure within the Channel Islands samples, $\Delta K$ suggested $K = 4$ as optimal, but the highest mean ln Pr(X | K) was for $K = 1$.

## Migration rate estimates

Our IMa2p analyses obtained an upper bound to the effective size of the Channel Islands population, but the analyses did not converge on an upper bound to the effective size of the mainland southern California population. This suggests that the effective size of

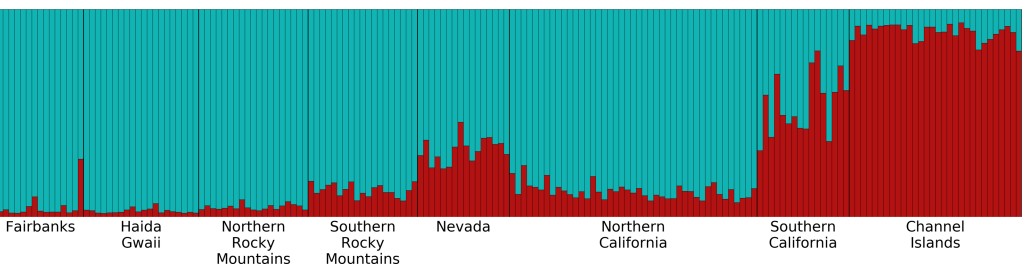

| Fairbanks | Haida Gwaii | Northern Rocky Mountains | Southern Rocky Mountains | Nevada | Northern California | Southern California | Channel Islands |

**Figure 5** **Structure plot for $K = 2$ with Channel Islands population included.** Different colors represent the two genetic clusters identified by Structure. Each vertical bar represents an individual *Oreothlypis celata*. The height of each color in a given bar illustrates the proportion of ancestry derived from each genetic cluster for that individual. When sample locality was excluded as a prior, we recovered $K = 1$.

the mainland population is likely much higher than that of the Channel Islands. Even though we were unable to accurately calculate migration rates scaled by population size, we were still able to assess the relative population sizes and rates of migration between the two populations. Across all three runs, we calculated a pairwise probability of 1.000 that the current effective population size of the southern California population is greater than that of the Channel Islands population. The probability that the current effective population size of the Channel Islands population is greater than that of the southern California population was <0.001. Our migration rate estimates were similar across our three IMa2p runs. Across all three runs, we estimated probabilities of 0.986 to 1.000 that the rate at which (looking forward in time) the southern California population receives genes from the Channel Islands population is greater than that of the reverse direction. Inversely, we calculated probabilities ranging from 0.000–0.013 that the rate at which (looking forward in time) the Channel Islands receives genes from southern California is greater than migration in the reverse direction.

## DISCUSSION

Genetic analyses of population structure in *Oreothlypis celata* revealed some structure in portions of the range and high levels of shared alleles across much of the mainland distribution of *O. celata*. The strongest result derived from our present dataset is that both the mitochondrial and microsatellite data suggested that the Channel Islands represent the most genetically distinct population included in our study. We found the highest genetic divergence between the Channel Islands and Fairbanks populations, the two most geographically distant populations in our analyses. More generally, the mitochondrial data suggested higher pairwise divergences among populations than the microsatellite data. The mitochondrial, but not the microsatellite data, supported statistically significant divergence between northern and southern *O. celata*. The microsatellite data provided weak support for isolation-by-distance across the species range.

Both the mtDNA and microsatellite data suggested that the Channel Islands *Oreothlypis celata* comprise a separate population that is distinct from the mainland population. Notable is the lack of *ND2* haplotype diversity (four haplotypes in 30 individuals with

27 individuals sharing the same haplotype) in the Channel Islands. This is suggestive of a founder event followed by persistence of a relatively small population that has likely fluctuated in size over time or of strong selection among mitochondrial genotypes to favor one genotype. Our mismatch distribution plots, Tajima's D and Fu's $F_S$ results, are consistent with the population recently having expanded in size. The nucleotide diversity within all other populations was much higher than that of the Channel Islands. Within the Channel Islands, the northern and southern island populations (represented by samples from Santa Cruz Island and Santa Catalina Island, respectively) did not display divergence in pairwise $F_{ST}$ or $\Phi_{ST}$ comparisons of the mtDNA gene *ND2* or in pairwise $F_{ST}$ or $R_{ST}$ comparisons of microsatellite data. Sequences from both islands clustered in our phylogenetic trees and haplotype network, suggesting that *O. c. sordida* from the northern and southern Channel Islands constitute one large population. The *O. c. sordida* individuals from Santa Cruz Island all shared the same *ND2* haplotype, which was also present on Santa Catalina Island. We identified three additional *ND2* haplotypes unique to Santa Catalina Island. The difference in haplotype diversity could be merely a sampling artifact, but this is unlikely given our sample size of 15 individuals from each island. Although the northern and southern Channel Islands may, in fact, be two separate populations that are diverging, any divergence is likely too recent to be statistically detected with our genetic data, despite the high mutation rates of our markers.

In contrast with other subspecies of *Oreothlypis celata*, *O. c. sordida* from the Channel Islands do not undertake a lengthy migration, although individuals may move short distances outside of the breeding season (*Gilbert, Sogge & Van Riper III, 2010*). The non-migratory tendency of *O. c. sordida*, its geographic isolation on the Channel Islands, and the smaller population size on the islands compared to the mainland have all likely contributed to the genetic differentiation that we observed. Interestingly, there also appears to have been cultural evolution of *O. c. sordida* on the Channel Islands, as evidenced by its slightly slower and more patterned songs compared to more rapid, less patterned songs of the nearest populations of *O. c. lutescens* (*Dunn & Garrett, 1997*). Based on the distinct phenotypes of island *O. c. sordida* individuals, *Johnson (1972)* hypothesized that the Channel Islands *O. celata* populations have been isolated from the mainland for a substantial period of time. The low degree of divergence and diversity in the mitochondrial data and the paucity of private microsatellite alleles do not support his hypothesis; rather, they suggest that the phenotypic differences in the *O. c. sordida* populations are of relatively recently derivation.

We obtained evidence for significantly greater gene flow from the Channel Islands to mainland southern California than in the reverse direction, a pattern that also has been detected in horned larks (*Eremophila alpestris*: (*Mason et al., 2014*). Both mitochondrial and microsatellite data supported *O. c. sordida* being more closely allied to coastal *O. c. lutescens* populations than to those of the interior *O. c. orestera*, contradicting *Johnson (1972)* hypothesis of a closer relationship between *O. c. orestera* and *O. c. sordida*. However, the Structure analysis in which we excluded the Channel Islands population suggested similar ancestry in the *O. c. lutescens* population of mainland southern California and the *O. c. orestera* populations of the Southern Rocky Mountains and Nevada (Fig. S4).

Of the four *Oreothlypis celata* subspecies, our molecular data most strongly supported *O. c. sordida* from the Channel Islands as a distinct group. Although *O. c. sordida* occurs primarily on the islands, it also breeds locally along the coast of mainland southern California (*Unitt, 1984*; *Dunn & Garrett, 1997*) in close proximity to *O. c. lutescens*. This distributional pattern is consistent with our finding of greater gene flow from the Channel Islands to mainland southern California than from the mainland to the islands. Recent expansion of the breeding range of *O. c. lutescens,* especially southward in San Diego County, has closed the distributional gap mapped by *Grinnell & Miller (1944)* between these two subspecies and caused some to suggest that *O. c. lutescens* is swamping out *O. c. sordida* on the mainland (*Unitt, 2004*). Further study combining specimens in known breeding condition with molecular markers is needed to test this hypothesis.

Although our microsatellite data showed statistically significant pairwise divergences between all pairs of subspecies except between *O. c. lutescens* and *O. c. celata*, our other methods did not recover genetic clusters that clearly distinguished subspecies other than *O. c. sordida*. Ongoing gene flow between *O. celata* subspecies may be acting to prevent greater divergence of populations. Using microsatellite data, *Bull et al. (2010)* calculated significant gene flow from populations of *O. c. lutescens* into *O. c. celata*. *Gilbert & West (2015)* provided further evidence of gene flow between these two subspecies by identifying *O. celata* individuals from Alaska that were morphologically intermediate between *O. c. celata* and *O. c. lutescens*.

## CONCLUSIONS

Overall, our results suggest that the differentiation seen in phenotypic and ecologic characters across *O. celata* is recent. Similar to the findings of *Bull et al. (2010)* for northern populations of *O. c. celata* and *O. c. lutescens*, genetic distances and clusters we observed across the western North American range of *O. celata* are consistent with high levels of gene flow combined with weak isolation-by-distance. Moreover, our finding that the strongest signal of population divergence occurs on the Channel Islands is consistent with geographic isolation, reduced migration tendency, and relatively low levels of gene flow from the mainland to the islands. The observation that cultural evolution in songs of *O. celata* has occurred on the Channel Islands (*Dunn & Garrett, 1997*) also supports the distinctiveness of this taxon on the islands. Future research that includes vocal as well as genomic data will further advance our understanding of the origin and evolution of birds on the Channel Islands. In summary, island isolation, subspecies (delineation by morphology, ecological, and life-history characteristics), and isolation-by-distance, in that order, are the likely best explanatory variables of the geographic structure we detected across the range of *O. celata*.

## ACKNOWLEDGEMENTS

We thank Anand Varma for permission to reproduce the photos included as Figs. S1 and S2. For access to specimens and genetic samples, we thank Andrew Johnson, Alison Boyer, and the Museum of Southwestern Biology; Kevin Burns and the San Diego State University

Museum of Biodiversity; Vicki Friesen, Kimberley Lemmen, Scott Taylor, Theresa Burg, and Roger Bull of Queen's University and the Canadian Museum of Nature; Jocelyn Hudon and the Royal Alberta Museum; Darren Irwin, David Toews, Alan Brelsford of the Department of Zoology and the Biodiversity Research Centre, University of British Columbia, Vancouver; and the Museum of Vertebrate Zoology. For assistance in the field, we thank Jessica Castillo and Janette Havens. We appreciate support provided by Mark R. Stromberg in our collection activities at Hastings Natural History Reservation. We thank Monica J. Albe for help with specimen preparations and field equipment, Lydia Smith for laboratory support, Michelle Koo for assistance with graphics construction, and Anna Sellas for advice regarding laboratory work and data analyses. Kimball Garrett and two anonymous reviewers provided helpful comments that improved an earlier version of this manuscript.

### Funding

Funds provided by the Sponsored Projects for Undergraduate Research Program (SPUR), College of Natural Resources, University of California, Berkeley (to Zachary R. Hanna) made this work possible. The funders had no role in study design, data collection and analysis, decision to publish, or preparation of the manuscript.

### Grant Disclosures

The following grant information was disclosed by the authors:
Undergraduate Research Program (SPUR).
College of Natural Resources, University of California, Berkeley.

### Competing Interests

The authors declare there are no competing interests.

### Author Contributions

- Zachary R. Hanna performed the experiments, analyzed the data, contributed reagents/materials/analysis tools, prepared figures and/or tables, authored or reviewed drafts of the paper, approved the final draft.
- Carla Cicero conceived and designed the experiments, contributed reagents/materials/-analysis tools, authored or reviewed drafts of the paper, approved the final draft.
- Rauri C.K. Bowie conceived and designed the experiments, performed the experiments, analyzed the data, contributed reagents/materials/analysis tools, prepared figures and/or tables, authored or reviewed drafts of the paper, approved the final draft.

### Animal Ethics

The following information was supplied relating to ethical approvals (i.e., approving body and any reference numbers):

UC Berkeley Animal Care and Use Committee provided approval for sample collection and this research under Animal Use Protocols R285 and R317.

## Field Study Permissions

The following information was supplied relating to field study approvals (i.e., approving body and any reference numbers):

We collected samples under California Department of Fish and Game scientific collecting permit numbers SC-458 and SC-10109 as well as US Fish and Wildlife Service permit number MB153526.

## DNA Deposition

The following information was supplied regarding the deposition of DNA sequences:

The ND2 sequences are available from NCBI (GenBank accession numbers MG686636–MG686821 and MH543328–MH543332; Table S1). The microsatellite genotypes are available in Table S4.

## Data Availability

We provide further methodology details and scripts on GitHub (https://github.com/zacharyhanna/ocwa-popgen).

## Supplemental Information

Supplemental information for this article can be found online at http://dx.doi.org/10.7717/peerj.7388#supplemental-information.

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
