# Peer review of "Molecular evidence that the Channel Islands populations of the orange-crowned warbler (Oreothlypis celata; Aves: Passeriformes: Parulidae) represent a distinct evolutionary lineage"

_PeerJ, doi:10.7717/peerj.7388_

## Round 0.1 · original submission · Major Revisions

Dear authors

Your ms has been reviewed and the recommendation was between minor and major revision. Please read the comments carefully and try to change your ms accordingly. Be aware that we will send your revision to the same reviewers.

Kind regards
Michael Wink
AE

Reviewer 1 ·

Basic reporting

no comment

Experimental design

no comment

Validity of the findings

no comment

Additional comments

I really enjoyed reading Hanna et al. The paper is very well written and is an important contribution to the phylogeographic and ornithological literature. I especially appreciated the detailed and thorough descriptions of the methods, because the genetic data did not have clear phylogeographic breaks in the form of clades the authors were forced to examine several populations groupings. This required detailed descriptions of how the samples were grouped and the authors should be proud that these descriptions were clear and concise. I found them very easy to follow and made the paper pleasant to read.

I believe the paper is acceptable for publication in PeerJ. There is little to critique with the manuscript overall. The figures and tables are all necessary and easy to read. The methods clear and the data generated done so following proper animal care and use guidelines and plans made to make datasets etc publically available. My only overall comment concerns the discussion. There are a few confusing comments made about incomplete lineage sorting and gene flow. This is the only part of the paper where I believe the authors get tripped up. It appears at the beginning of the discussion they make the argument that lineage sorting is responsible for shared variation among populations but later on they contradict this statement and suggest it is geneflow. I just think these ideas need to be reconciled and it wouldn’t require a major revision to do so. Below I point to the specific instances of discord in the text.

Line 421-423: I am not convinced that the lack of structure and high levels of shared alleles is solely due to incomplete lineage sorting. Seems to me that there is just as good of a chance that there is high gene flow that flows through adjacent populations which is why you see a pretty strong signal of IBD (the most distant populations being the most diverged). I would just be cautious with this statement.

Line 432-435: I understand your reasoning that the low diversity might be indicative of a founder event, bottleneck or strong selection, but your mismatch and D and F stats don’t appear to back up this conclusion. So what makes you lean towards such events. Low diversity on islands can also come from persistent stable but small populations.

Line476-477: This statement contradicts you statement in the opening paragraph that suggest incomplete lineage sorting is the process behind shared variation.

Minor recommended edits:

Line 40: delete “and” at the end of the sentence.

·

Basic reporting

The paper is well-written, though as in all such papers the molecular laboratory methodology and bioinformatics analyses are essentially impossible to present in a fashion readable and intelligible to the non-specialist (note that this is not a generic problem, not a failure of this particular contribution). The general topics, results and interpretations are presented clearly.

In setting up the study (lines 26-28), it might be worth mentioning that there is geographical variation in songs of Oreothlypis celata, and the slightly slower and more patterned songs of sordida (in contrast to more rapid, less patterned songs of the nearest populations of lutescens) have been commented upon by some authors. In the absence, however, of well-designed, quantitative studies of song variation, perhaps there is nothing to site here beyond some qualitative mention in popular literature (Dunn and Garrett 1997). Perhaps also worth mentioning that sordida is the only subspecies that regularly nests well above ground level (see BNA account). The habitat term "riparian chaparral" (line 31) is a bit odd, but certainly lutescens does breed commonly in the chaparral/riparian ecotone. Line 39 (breeding range of orestera) might better say "and in some mountain ranges within the western deserts of North America" so as not to imply they nest in desert habitats; also, that sentence seems to be truncated in my version.

Experimental design

One very minor comment: Regarding population sampling (lines 66-74), bear in mind that some southern California coastal breeding Orange-crowned Warblers can disperse well away from breeding areas by mid to late May (e.g. up into the higher mountains or even onto the deserts), so you might point out how you took care to be certain your samples came from breeding sites and did not include dispersing post-breeders.

Validity of the findings

Regarding your population #7: It is unfortunate that there were no available specimens and no realistic collecting opportunities in places like the Santa Monica Mtns. in Los Angeles/Ventura Counties where lutescens nests in close proximity to mainland populations of sordida (the latter well-known from the Palos Verdes Peninsula but probably locally a bit farther up the coast, and in several populations down the coast in Orange and San Diego Counties). You might look at Philip Unitt's comments in the San Diego Bird Atlas on the status of sordida in San Diego County, the southward range expansion of lutescens there, and the possibility that lutescens if "swamping out" sordida in costal San Diego.

Additional comments

A good contribution that does not exactly argue that current subspecies delineation is flawed (insofar as the subspecies concept is informative, as I believe it to be), but does demonstrate that the underlying genetics are far more complex. Your conclusions of the distinctiveness of sordida will hopefully spur more work on the ecology, behavior and phenotypic distinctness of this taxon, as well as a deeper investigation of the relationship between coastal mainland breeding populations of lutescens from Santa Barbara and Ventura Counties south to San Diego County and the localized mainland breeding populations of sordida.

Reviewer 3 ·

Basic reporting

Mostly fine.

Experimental design

This is good -- strong geographic sampling in western N America; mtDNA and nuDNA.

Validity of the findings

I thing IBD is overemphasized given the overall results; more suggestions below.

Additional comments

PeerJ #31580. Hanna et al. on orange-crowned warbler population genetics.

The authors used mtDNA and 10 microsatellite loci to assess genetic properties of the orange-crowned warbler across part of its range (focusing on western North America), including all four recognized subspecies and 192 individuals. The sampling represents a considerable degree of field effort over many years. Overall, there were low levels of divergence, attributed in the abstract to IBD (though I would change; see below). Channel Islands populations are focused on as being most distinct.

Overall, the dataset, analyses, and writing are fairly sound. I do have some suggestions for improvements, however. With the infusion of some arbitrariness in making geographic assignments, the geographic relationships and differences are enhanced. This results I think in needless de-emphasis of insular isolation and phenotypic attributes (subspecies), which seem to be much stronger influences on patterns in the data.

The paper focuses rather heavily on the second of four arrangements of geographic analyses (Figs. 1, 2b[needs to be labeled that], and 3). This arrangement partitions the samples (incompletely) into 8 populations (Fig. 1, lines 75-78). Designation of some of these populations and omissions of some of the sampled localities (Fig. 1, NW Canada) seems to have a great degree of subjectivity. The omission of 7 disparate points in NW Canada and the island-mainland separation of populations 7 & 8 would bias the subsequent analysis. You can see the bias this would introduce to enhance different signal in the data: eliminating samples in NW Canada that are similar across great distances and emphasizing the islands (dissimilar over short distances) as separate populations despite proximity. Ultimately, there were four groupings considered, but the first, with only locality and with “weak” evidence (line 387) for geographic distance being important is unduly de-emphasized. I am not objecting to multiple approaches, because it is useful to explore what factors best explain the distribution of genetic variation across this space, but a key result remains “these initial analyses did not reveal spatial structure in the genetic data” (line76), and the paragraph beginnng on line 384. The overall approach used can appear as being “let’s throw things at the wall until something sticks,” and in Methods at least it might be delivered better. Currently it reads like “Well, straight-up raw analyses didn’t show anything interesting, so we tried these other things...”

I am a skeptic of using locality priors in Structure analyses. Its effect is to impose some structure where it might not exist on a genetic basis alone. That is problematic in organisms with large dispersal distances. With these animals being seasonal migrants with imperfect site fidelity and probably long dispersal distances, pure analyses of the underlying genetic properties are more informative. So Figure 3 should not be the Structure result that takes priority. The first Structure analysis that includes all samples and no locality information (line 391, K = 1) should be at a minimum 3a. That is much more comparable to most published Structure analyses in the literature. The legend also needs to include what data were used and whether priors were used.

Despite there being a *lot* of statistical tests in Tables 1-4, S2-S4, there is little evidence of multiple test corrections (line 334 mentions Bonferroni corrections outside tabulated results). This has become less acceptable, and it is where the penalty is usually paid for a “let’s see what sticks” approach. Careful, methodical planning between rounds of tests is usually evident to balance these tradeoffs. This should be addressed. It is partly dealt with in language in the text looking for consensus patterns among analyses, but more is needed.

The gene flow analyses seem incomplete. I have not used IMa2p with mtDNA and microsatellies, but most analyses on this topic provide an actual rate estimate of individuals per generation and effective population size estimates, not just that gene flow is asymmetrical and that population sizes differ in the expected direction. Traditional analyses (which the authors were not shy to use elsewhere in the study) should be used to analyze mtDNA and microsatellite data separately to provide effective populations size and gene flow estimates (which can then be directly compared to a vast literature on these topics in population genetics). Using more traditional IM analysis on mtDNA could also provide a decent divergence estimate and improve the authors’ suggestion in the Discussion (lines 453-459) that divergence is more recent than previously suggested.

The ending seems really limp to me, and actually misdirecting from the strongest takeaway results. IBD was not supported with mtDNA and it was remarkably weak with microsatellites. On the other hand, insular isolation on the Channel Islands produced the strongest signal of population divergence, and the significant differences among subspecies (Table S2) were also quite remarkable. How is this consistent with IBD “having generated the genetic distances and clusters we observed”? This overemphasizes the IBD signal and underemphasizes other aspects of the data. To me, the results indicate that life history (lower migration) plus island isolation and then, among other subspecies, ecological/life history aspects not considered are more important drivers of the distribution of genetic variation in this species across this space. In fact, an a priori prediction would have probably ranked island isolation, subspecies, and distance in that order as the likely best explanatory variables. If the authors think not, a more convincing case should be made for the aproach they’ve taken to emphasize distance alone. For a start, I’d move Fig. S3 and Table S2 into the main body of the paper, then provide a more robust consideration of all the factors to explain the distribution of genetic varaition in the data. (And did I miss an AMOVA explanation of the amount of variation explained by subspecies?).

Other notes made:

Line 302: “...we recovered significant isolation by distance in the microsatellite data” is contrary to lines 76-77 (“did not reveal spatial structure”). In fact, the raw microsatellite results in showing such a poor relationship (lines 386-390) corroborate the SAMOVA result here: IBD is not an important component of these populations. I think this needs emphasis.

Line 422: Why focus only on ILS? Gene flow is probably far more important for the “high level of shared alleles.”

The supplementary tables need legends in the files, including what the values are, what the data are, and the singificance levels of the asterisks.

---

## Round 0.2 · Minor Revisions

Dear authors
Your ms needs a final revision. Please follow the recommendations of reviewer 3 carefully.
Kind regards
Michael Wink

Reviewer 1 ·

Basic reporting

no comment

Experimental design

no comment

Validity of the findings

no comment

Additional comments

I have no further comments. The authors have fully addressed my previous comments and concerns. Nice work.

·

Basic reporting

All of the minor concerns I expressed in the initial round of review have been adequately addressed by the authors.

Experimental design

No comment

Validity of the findings

no comment

Additional comments

Again, all of the minor concerns I expressed in the initial round of review have been adequately addressed by the authors. These concern some details on breeding seasonality, potential issues with early post-breeding dispersal, possible vocal differentiation of sordida, and areas where sordida and lutescens breed in close proximity on the mainland. I reiterate that I am not qualified to assess the techniques and conclusions of the molecular work.

Reviewer 3 ·

Basic reporting

See all comments in comments to the authors.

Experimental design

See all comments in comments to the authors.

Validity of the findings

See all comments in comments to the authors.

Additional comments

Hanna et al. 2019 PeerJ population genetics of orange-crowned warbler

I saw this paper before and see many improvements following from my previous comments. It is a good study, and publishable. But there are still two places where I think the authors could improve the overall impacts of their paper.

The first response does not address my point. Reviewer 1 commented on the clarity of the grouping treatments, not on the appropriateness of those groupings nor on how the analyses were subsequently treated. I did not critique their explanation, but rather their subsequent treatment (and a stylistic approach I would not choose myself). De-emphasizing the unpartitioned results (i.e., “did not reveal spatial structure” line 104) was I thought somewhat misleading. “We recovered low levels of divergence...” in the Abstract I think minimizes the results summarized in line 104 and overemphasizes the post-partitioning results, which masks an important result of your study. Using geography alone, no significant structure was found. However, with geographic partitioning that took into account, e.g., “ecologically significant” landmarks (line 112), and subspecies (line 119), significant levels of structure/divergence were recovered in your data. Styllistically, I would have made the paritioning seem less subjective and more directly in pursuit of factors likely to cause genetic structure, but differences there are valid choices.

Thank you for pointing out the multiple-test corrections. It looks like I did not look carefully at the table legends given on separate pages from the tables themselves per an unfortunate journal style (that should be changed, please, editors).

I also appreciate minimizing issues of IBD and ILS. That helps. The new Table 4 does, too.

In the response, I am surprised by this statement:“...we were unable to effectively calculate migration rates scaled by population size...” It needs to be clear that you have *chosen* not to effectively calculate these by not doing analyses that have been routine in mtDNA and microsatellite studies.

Contrary to your opinion, IM is well suited to analyzing mtDNA for population genetics. In fact, Jody Hey et al’s program IM has been used in hundreds of studies doing exactly what I suggested you do (see, e.g., Hey 2005 PLoS Biology and the citations of it). MtDNA is nonrecombining, which is a strength for this analysis, and the results apply only to females (e.g., effective population size estimates). Your results would not only go directly to answering some key questions in your study (date of split, effective population sizes, rates of gene flow), they would be directly comparable to these hundreds of similar studies. Unlike IMa2, it does not require a supercomputer.

Similarly, your microsatellite data provide a genomically independent, complementary basis for evaluating gene flow. Arguably the best direct approach would be to use Beerli & Felsentein’s 1999 (software updated through 2012) program migrate.

So I still see an easily filled gap in your objectives given your data to provide estimates not of relative rates, but of actual individuals per generation (using two great datasets!). Because you are right: “Furthermore, comparisons of island taxa and their mainland counterparts are fundamental to assessing the taxonomic status of island endemics, many of which are of conservation concern (Wilson et al., 2009).” (First paragraph of Introduction). No ddRAD data are needed to do this, and it’s been routinely done in literally hundreds of other studies like yours. Fringe benefits include actual estimates of effective population sizes and divergence dates for the Channel Islands population.

I think it is a good study. But it would be fairly easy to make it a great study, given the data you already have in hand.

---

## Round 0.3 · accepted · Accept

Dear authors

Good news. Your revision is acceptable. Thank you for publishing in PeerJ

Regards
Michael Wink
AE